# Male-Type Depression Mediates the Relationship between Avoidant Coping and Suicidal Ideation in Men

**DOI:** 10.3390/ijerph191710874

**Published:** 2022-08-31

**Authors:** Kieran M. O’Gorman, Michael J. Wilson, Zac E. Seidler, Derek English, Ian T. Zajac, Krista S. Fisher, Simon M. Rice

**Affiliations:** 1Orygen, Parkville, VIC 3052, Australia; 2Centre for Youth Mental Health, The University of Melbourne, Parkville, VIC 3010, Australia; 3Movember, Richmond, VIC 3121, Australia; 4Commonwealth Scientific and Industrial Research Organisation (CSIRO), Adelaide, SA 5000, Australia; 5School of Psychology, University of Adelaide, Adelaide, SA 5000, Australia

**Keywords:** avoidant coping, suicide, self-harm, male-type depression, COVID-19, substance use, emotion suppression, anger, aggression

## Abstract

Despite known links between men’s avoidant coping behaviours (e.g., distraction, denial, substance use) and suicide risk, little research has explored the mechanisms underpinning this relationship. This study sought to examine whether male-type depression symptoms (e.g., anger, aggression, emotion suppression), assessed by the Male Depression Risk Scale, mediate the association between avoidant coping and suicide/self-harm ideation in men. Data were drawn from an online survey of a community sample of 606 Australian men (*M* age = 50.11 years; SD = 15.00), conducted during the COVID-19 pandemic. Mediation analyses were applied to examine the effect of male-type depression on the association between avoidant coping and suicidal/self-harm ideation, controlling for age, resilience and the experience of two psychosocial stressors during the COVID-19 pandemic (financial stress and government restrictions). Avoidant coping was associated with suicidal/self-harm ideation, *r* = 0.45, *p* < 0.001. Results supported a mediating role of male-type depression symptoms in this relationship, *R*^2^
*=* 0.29, *P_M_* = 0.36, *p* < 0.001, underscoring the importance of screening for male-type depression symptoms to better identify men at risk of suicidal/self-harm ideation. Results also suggest a need to support men to develop effective coping strategies, particularly in the context of common psychosocial stressors experienced during the COVID-19 pandemic and beyond.

## 1. Introduction

Depression is the leading cause of disability worldwide [1] and is associated with poor educational attainment, low quality of life and early mortality [2]. Twice as many women are diagnosed with major depressive disorder compared to men [3], yet men are four times as likely to die by suicide [4] and exhibit significantly higher rates of alcohol and other drug use [5]. Amongst other factors, these disparities may be influenced by adherence to traditional masculine norms (e.g., self-reliance and stoicism) which discourage the expression of typical depressive symptoms (e.g., sadness) and proactive help-seeking in men, instead coinciding with maladaptive coping strategies (e.g., substance misuse) [6].

Coping strategies are emotional or behavioural regulation strategies people use to process and respond to psychological distress [7]. Avoidant coping is one such strategy identified in the Brief Coping Orientation of Problem Experience Inventory [8]. As the name suggests, avoidant coping strategies are utilized in order to avoid distressing thoughts or feelings associated with a stressor. Such strategies include substance use (using alcohol or drugs to counter the distressing feelings associated with the stressor), denial (refusing to acknowledge the stressor), behavioural disengagement (ceasing goal pursuits that are thwarted by the stressor), and mental disengagement (distracting oneself to avoid thinking about the stressor) [7]. Avoidant coping strategies are often considered maladaptive, as they are associated with poor mental health outcomes such as depression, deliberate self-harm (DSH), suicidal ideation, and suicide attempts [9,10,11,12,13,14,15]. Moreover, leveraging alcohol and other drug use to as a coping strategy is associated with DSH in men [16,17], highlighting a need to specifically explore the psychological and behavioural sequelae of avoidant coping strategies in men. In contrast, active coping strategies consist of attempting to resolve, understand, or reappraise a stressor [18]. These strategies are often considered adaptive as they are related to a greater sense of psychological wellbeing [19,20,21].

Another factor related to mental-ill health in men is their adherence to traditional masculine norms. Conformity to the traditional masculine norms of self-reliance and playboy (i.e., having multiple sexual partners) are both associated with greater depression severity in men [22,23], while conformity to the traditional masculine norms of violence and self-reliance are both associated with suicidal ideation [24,25]. Masculine norms also affect sexual minority men, such that gay and bisexual men who self-identify as more masculine experience greater depression in response to discrimination, relative to sexual minority men identifying as less masculine [26]. Standard screening measures such as the Patient Health Questionnaire [27] assess prototypical symptoms of depression (e.g., depressed mood, anhedonia and sleep/appetite changes), but depressed men may also exhibit externalizing symptoms, i.e., outward expressions of psychological distress such as anger, aggression, and substance use [28]. Men with depression are more likely than women to report symptoms including anger [29], risk-taking, and drug/alcohol use [30], which may be influenced by gender role socialization (i.e., the gendered ways in men are taught to think, act, and feel).

The Male Depression Risk Scale (MDRS-22) [31] was developed to measure these male-type depressive symptoms. The MDRS-22 comprises six externalizing symptom domains: (1) emotion suppression, (2) drug use, (3) alcohol use, (4) anger and aggression, (5) somatic symptoms, and (6) risk-taking. The scale has been validated in multiple countries [32,33,34] and older and younger age-groups [35]. The MDRS-22 shows convergent validity with the PHQ-9 but outperforms the PHQ-9 in identifying recent male suicide attempts [32]. The MDRS-22 may be particularly suited to identifying psychological distress in men who display uniquely externalizing symptoms (i.e., low PHQ-9 and high MDRS-22 scores), approximately one-third of whom are not identified as depressed by PHQ-9 cut-offs [36]. To allow for efficient administration, a seven-item version of the MDRS (MDRS-7) was recently validated [37]. The short version shows promising predictive validity: men who score highly on the MDRS-7 are at increased risk of mental illness and suicide risk [37].

To date, little research has investigated the mechanism of association between avoidant coping and suicidal/self-harm ideation, with no studies considering potential mediating effects of male-type depressive symptoms. However, there are some indications that men adhering to traditional masculine norms utilize less effective coping strategies, with men being less likely than women to employ effective coping strategies, such as seeking emotional support from others [38] and are more likely to misuse alcohol and drugs [30]. Men who conform to the traditional masculine norm of dominance and employ avoidant coping strategies report more severe depressive symptoms [23]. Masculine norms including self-reliance have been shown to discourage men from employing coping strategies (e.g., seeking mental health treatment) which may require emotional vulnerability [39]. Avoidant coping is associated with an increased risk of suicide in both genders [40], but men who experience multiple stressful life events report more severe male-type depressive symptoms [33], suggesting that men cope with such stressors in ways that give rise to different symptoms.

Avoidant coping and male-type depressive symptoms are also related at a conceptual level. Symptoms of anger and aggression, for example, may result secondary to feelings of sadness or shame and at times manifest in an externalized form congruent with traditional masculine norms [28]. The avoidant strategy of denial (e.g., denial that one is feeling depressed or denial of the reasons one is depressed) is also congruent with masculine norms which may label feelings of sadness as being effeminate.

Whilst avoidant coping strategies are generally considered maladaptive in the literature, it is important to note that coping styles and behaviours—in and of themselves—are generally not considered to be inherently adaptive or maladaptive [41]. Instead, the adaptiveness or ‘maladaptiveness’ of a coping behaviour might reflect the particular stressful context it is applied to and is further complicated by the often-dynamic nature of stressors; wherein an initially effective coping strategy might later become ineffective as the stressor or its demands change [42]. For example, Chao [43] showed in college students that more frequent use of avoidant coping behaviours predicted lower wellbeing when other coping behaviours were held constant. In other words, rigid adherence to a particular style of coping—such as avoidance—is more problematic than leveraging a range of different coping behaviours. Children who attempt to use active coping to cope with uncontrollable stressors such as parental divorce exhibit more severe externalizing symptoms and poorer social functioning than those who use active coping strategies to cope only with controllable stressors such as conflict with siblings [18]. Conversely, youth who use avoidant coping strategies to cope with extreme, uncontrollable stressors (e.g., severe trauma, homelessness, parental conflict) exhibit less severe post-traumatic, anxiety, and depressive symptoms [44,45,46].

Avoidant coping strategies may prove maladaptive in the context of stressors such as the COVID-19 global pandemic [47], given the reduced accessibility of protective factors including social support, physical activity and access to medical care [48] for conditions such as depression. Active coping strategies appear to have had a protective effect for men during the pandemic [49], despite the uncontrollable nature of many COVID-related stressors. In the face of such stressors, Giuntella and colleagues [48] found that resilience (i.e., the tendency to “bounce back” from stress) [50] had a greater effect as a protective factor for depression during the pandemic than prior. Therefore, both the international context (the pandemic) and the individual context (age and personal resilience) may be implicated in the relationships between avoidant coping, male-type depression, and suicide/self-harm risk.

As described, there are empirical and theoretical reasons to expect that male-type depressive symptoms may mediate the association between avoidant coping and suicidal/self-harm ideation. If the relationship between avoidant coping and suicidal/self-harm ideation is mediated by these symptoms, then a focus on avoidant coping is likely to be insufficient to identify suicide risk in depressed men. The present study aimed to (i) validate the factor structure of the MDRS-7 identified in Herreen and colleagues [37] using confirmatory factor analysis, and (ii) investigate whether the MDRS-7 mediates the relationship between avoidant coping and suicidal/self-harm ideation in a sample of Australian men. Given that the MDRS-7 assesses a range of symptoms, we also investigated the relationship of individual MDRS-7 items (e.g., ‘It was difficult to manage my anger’) to suicidal/self-harm ideation.

## 2. Materials and Methods

### 2.1. Participants and Procedure

Australian men aged ≥16 years were invited to take part in a brief online survey about their mental health and any help-seeking experiences during the COVID-19 pandemic. Location data were embedded in the survey such that any participants outside of Australia were automatically excluded. Data presented here are a subset of the larger survey that was hosted by Qualtrics. Recruitment occurred from 25 October to 29 December 2021, via targeted Facebook advertisements. Advertisements contained the following text, mirroring the approach applied in previous male-specific help-seeking surveys [51]: “*Survey for men: Have you had any difficulties with your mental health during the COVID-19 pandemic? We want to hear about your experience. Complete our short 10–15 min survey here*”.

Participants who clicked through the advertisement link were immediately presented with a plain language statement and consent form with a yes/no response prompt. Consenting participants were then asked to work through the ~15 min survey, which contained a range of quantitative and free text entry items exploring recent mental health and help-seeking experiences. Participants were given the option to enter the draw for a $500 voucher as compensation for their time. Ethical approval for the study was granted by the University of Melbourne Human Ethics Sub-Committee (ethics ID: 2021-13657-22724-4).

### 2.2. Measures

#### 2.2.1. Demographics

Participants reported their age, gender, place of residence, relationship status, sexual orientation, employment status, level of education, and income. Participants also reported whether they identified as Aboriginal or Torres Strait Islander, and whether they identified as transgender. See Appendix A for information regarding the measurement of demographics.

#### 2.2.2. Avoidant Coping

The Avoidant Coping subscale of the Brief COPE inventory [8] was used to assess coping facets of self-distraction, substance use, denial, and behavioural disengagement. The Avoidant Coping subscale [52,53] is composed of 8-items (e.g., ‘*I’ve been turning to work or other activities to take my mind off things’*), using a scale of 1 (I haven’t been doing this at all) to 4 (I’ve been doing this a lot). Higher scores reflect greater use of avoidant coping. Cronbach’s alpha in this sample was 0.671.

#### 2.2.3. Male Depression Symptoms

Male depressive symptoms were assessed via the recently validated MDRS-7 [37] in Australia. The MDRS-7 has a simplified Likert response scale relative to the original version. It assesses six dimensions of male-type depressive symptoms refined from the longer form MDRS-22 [30]: (1) emotion suppression, (2) drug use, (3) alcohol use, (4) anger and aggression, (5) somatic symptoms, and (6) risk-taking. Each dimension is measured by a single item except for anger and aggression, measured by two separate items. Respondents rate items (e.g., ‘*I bottled up my negative feelings*’) relative to the preceding month on a scale from 0 (none of the time) to 4 (all of the time). Higher scores indicate higher levels of depression. Cronbach’s alpha in this sample was 0.778.

#### 2.2.4. Resilience

The Brief Resilience Scale (BRS) [50] was used to measure resilience and is composed of 6 items rated on a scale of 1 ‘strongly disagree’ to 5 ‘strongly agree’. Items include *‘It does not take me long to recover from a stressful event’* and *‘I tend to take a long time to get over set-backs in my life’* (reverse-coded). Higher scores indicate higher levels of resilience. Cronbach’s alpha in this sample was 0.869.

#### 2.2.5. Suicidal and Self-Harm Ideation

Suicidal and self-harm ideation was measured using item 9 of the PHQ-9 [27], which asks respondents to rate the frequency with which they have had ‘*Thoughts that you would be better off dead or hurting yourself in some way’* over the preceding two weeks, with responses ranging from 0 (‘*Not at all’*) to 3 (‘*Nearly every day’*). For the exploratory penalised logistic regression (Section 3.4), suicide/self-harm ideation was treated as binary to test the clinical utility male-type depressive symptoms, i.e., their ability to distinguish men experiencing *any* suicidal ideation from those experiencing *none*. Responses ≥1 were taken to indicate recent suicidal or self-harm ideation.

#### 2.2.6. COVID-19 Stressors

Two questions adapted from Ogrodniczuk et al. [54] were used to assess the psychosocial impact of COVID-19. Participants responded on a Likert scale to the questions ‘*To what extent has the COVID-19 pandemic put financial stress on you?’* (Ranging from 1= No stress to 5= Extreme Stress) and ‘*How have government COVID-19 restrictions affected your mental health?’* (Ranging from 1 = Very Positively to 5 = Very Negatively). As such, higher mean scores are indicative of higher levels of stress or negative experience. Three additional questions used by Ogrodniczuk et al. [54] (regarding job loss, relationship impacts, and changes in alcohol consumption habits) were not included in this study because their response scales were categorical and so could not be included in our analysis which required continuous variables. 

#### 2.2.7. Help-Seeking

Two questions were used to assess participants’ help-seeking during and prior to the COVID-19 pandemic. Participants were asked “*Have you sought help from a mental health professional (i.e., psychologist, counsellor, or other therapist) since March 2020? i.e., during the COVID-19 pandemic?*”, to which they could respond “Yes” or “No”. Participants were also asked “*Have you ever sought help from a mental health professional (i.e., psychologist, counsellor, or other therapist)? i.e., prior to the pandemic*”, to which they could respond “Yes—I have sought help but not in COVID-19 times” or “No—I have never sought help from a mental health professional”.

### 2.3. Data Analysis

Analyses were completed in IBM SPSS Statistics Ver 26 and Stata 15.0. Descriptive statistics characterised the sample and scale reliability coefficients were identified. Eight-hundred and nine individuals responded to the survey. Six-hundred and six male respondents (74.9% of the 809 respondents; mean age 50.11 years, *SD* = 15.00) completed all measures, while others dropped out progressively through the survey. Participants were included in each separate analysis only if they completed all the relevant measures (e.g., for the exploratory penalised regression, only participants who completed the MDRS-7 *and* the PHQ-9 were included). The response rate for the MDRS-7, presented early in the survey, was 87.6% (*n* = 706). 84.2% (*n* = 681) of respondents completed the PHQ-9, 78.1% (*n* = 632) presented the Brief COPE, 75.2% (*n* = 608) completed the BRS, and 74.9% (*n* = 606) completed the COVID stressor questions. 

Confirmatory factor analysis (CFA) was conducted for the MDRS-7 to validate the single-factor structure in the present sample; and anger and aggression items were permitted to covary. Fit indices reported include the comparative fit index (CFI), the Tucker–Lewis index (TLI); the root mean square error of approximation (RMSEA), and the standardized root mean residual (SRMR). Interpretation of these indices were guided by the recommendations of Hu and Bentler [55], wherein TLI and CFI values >0.95., RMSEA values <0.06, and SRMR values <0.08 are considered indicative of good fit.

Using the SPSS macro PROCESS 3.4 [56], a mediation model was utilized to evaluate the effects of the MDRS-7 total score (*M*) on associations between avoidant coping (*X*) and recent suicide and/or self-harm ideation (*Y*). PROCESS implements a non-parametric bootstrapping procedure free of assumptions of normality, equality of variances, etc. We utilized PROCESS model 4, with 99% CIs and 5000 resamples. Participant age, resilience responses on the BRS, and the two COVID-19 related stressors were entered as covariates. In order to investigate whether any significant results were simply an artefact of the use of multiple covariates [57], the mediation model was also conducted without covariates (i.e., only *X*, *M*, and *Y* variables). The effect size measure *P_M_* [58] was calculated to assess the magnitude of the indirect effect as a proportion of the total effect. *R*^2^ was reported as an indicator of model fit.

Finally, we conducted an exploratory penalised (Firth) logistic regression model to identify individual MDRS-7 items associated with recent suicide and self-harm ideation. Penalised logistic regression is the preferred logistic regression approach for unbalanced designs, evident when modelling lower likelihood events like suicide and self-harm ideation. Wald χ^2^ and Nagelkerke *R*^2^ values were reported as indictors of model fit, alongside adjusted odds ratios (AORs). 

## 3. Results

The sample was drawn from a larger pool of 809 men, nine of whom identified as transgender, with a mean age of 50.29 years (*SD* = 15.43). Most respondents were from metropolitan areas (64.5%, *n* = 522), with 28.4% (*n* = 230) from regional and 7.0% (*n* = 57) from rural or remote areas. Just under half of respondents indicated full-time employment (49.1%, *n* = 397), 20.9% were retired (*n* = 169), and 9.8% (*n* = 80) were unemployed. Most respondents were born in Australia (77.6%, *n* = 628), and 2.2% (*n* = 18) of respondents identified as Aboriginal or Torres Strait Islander. Most respondents (70.6%, *n* = 571) identified as straight, while 21.4% (*n* = 173) and 6.2% (*n* = 50) identified as gay or bisexual, respectively. Almost half (46.2%, *n* = 374) of respondents were married or in a de facto relationship. Most commonly, respondents lived with their partner but no children (30.7%, *n* = 248), while 23.7% (*n* = 192) were single and living alone. Regarding help-seeking, 27.2% (*n* = 220) of participants had sought help for mental health problems prior to the pandemic, while 48.9% (*n* = 396) of participants reported seeking help during the pandemic.

### 3.1. Scale Structure and Associations

Inspection of skewness and kurtosis values (all <1) indicated all outcome variables were normally distributed. CFA indicated that the single factor MDRS-7 model was an excellent fit to the data, with all fit indices exceeding recommended cut-offs; χ2(13) = 54.21, *p* < 0.001; CFI = 0.971; TLI = 0.953; RMSEA = 0.067; and SRMR = 0.036. Standardised item loadings are presented in Table 1.

Pearson correlations indicated significant associations (*p*’s < 0.001; see Table 2) between study variables. Generally speaking, the MDRS-7 and avoidant coping scores were moderately positively correlated with suicidal/self-harm ideation, and weakly inversely correlated with resilience. The MDRS-7 was positively associated with avoidant coping.

### 3.2. COVID-19 Impacts

Most (58.1%, *n* = 352) respondents indicated that the pandemic had placed at least some financial stress on them, with 12.7% (*n* = 77) and 9.1% (*n* = 55) reporting this stress was considerable and extreme, respectively. Two-thirds of respondents (66.7%, *n* = 404) reported that government pandemic restrictions had negative effects on their mental health (66.6%). The response distribution to the COVID-19 questions is detailed in Table 3.

### 3.3. Mediation Analysis

The bootstrapped mediation analysis predicting suicidal/self-harm ideation was significant, *F*(6, 599) = 41.147, *R^2^* = 0.29, *p* < 0.001. Results are presented as standardised regression coefficients (*β*) to allow for the effects of different scales to be compared. Avoidant coping had a significant direct effect on suicidal/self-harm ideation (*β* = 0.20, *p* < 0.001). MDRS-7 total scores positively mediated the relationship between men’s avoidant coping and recent suicidal/self-harm ideation (*β* = 0.11, 99% CI [0.01, 0.04], *P_M_* = 0.36), accounting for 36% of the total effect. Resilience was the only significant negative covariate (*β* = −0.20, *p* < 0.001). The full mediation results are presented in Table 4. The mediation model and indirect effect remained significant without covariates, *F*(2, 603) = 95.918, *R^2^* = 0.24, *p* < 0.001 (Appendix A), indicating that significant results were not merely due to the inclusion of multiple covariates.

### 3.4. Exploratory Penalised Logistic Regression

In order to examine the individual MDRS-7 items that were predictive of recent suicide and/or self-harm ideation, we examined MDRS-7 items relative to PHQ-9 item 9 (‘*Thoughts that you would be better off dead or hurting yourself in some way’;* dichotomised; no/yes), using penalised (Firth) logistic regression. Of the seven MDRS-7 items, three were significant predictors (*p* < 0.05) in the logistic model (Wald χ2(7) = 128.25, *p* < 0.001, Nagelkerke *R*^2^ = 0.338). The significant predictors, and their adjusted-odds ratios are presented in Table 5.

## 4. Discussion

The present study supports the psychometric validity of the MDRS-7 as previously established by Herreen and colleagues [37]. These findings extend insights regarding the MDRS-7 and its relationship to men’s suicide/self-harm risk and validate it’s use in the COVID-19 context. Principally, findings indicated that male-type depression symptoms mediated the association between avoidant coping and men’s recent suicidal/self-harm ideation. These effects held whilst accounting for the effects of participant age, resilience, and experience of COVID-19 related stressors. Exploratory analyses also indicated that the emotional suppression and substance use items from the MDRS-7 predicted recent suicidal/self-harm ideation. 

Findings of a positive association between avoidant coping and suicidal ideation among men mirror past research reporting the often-negative impacts of avoidant coping in relation to mental health [12,13,22]. Livingston and colleagues [49] used latent profile analysis to assess men’s use of various coping strategies both before and during the COVID-19 pandemic. They identified three groups of men: Relaxed Copers, who used relatively few coping strategies but primarily relied on acceptance and self-distraction; Approach Copers, who primarily relied on active coping strategies; and Dual Copers, who applied predominately avoidant coping strategies (e.g., self-distraction) with some active coping strategies (e.g., planning) [49]. Compared to Relaxed Copers and Approach Copers, Dual Copers were at greater risk of stress, depression, anxiety, and anger symptoms, alongside adopting more negative appraisals of pandemic-era stressors such as the loss of employment opportunities [49]. Whilst evidence suggests suicide rates have not risen in the context of the pandemic [59], scholars have anticipated the negative mental health impacts of global crises tend to peak following the passing of the initial event [60,61]. Men demonstrated flexibility in their use of coping strategies during the pandemic, such as by using fewer avoidant strategies and more active strategies, with largely positive effects on mental-wellbeing [49]. In conjunction with our finding that avoidant coping confers a risk for suicidal/self-harm ideation in men, this suggests that encouraging men to use fewer avoidant coping strategies and more active coping strategies may reduce their risk of depression and suicide.

Findings that male-type depression symptoms mediated the link between avoidant coping and suicidal ideation in men (accounting for approximately one-third of the total effect) further support conclusions regarding the unhelpful nature of avoidant coping processes in some contexts. Given that male-type depression is more common among men who strictly adhere to traditional norms of masculinity [23,31], it may be the case that their socialisation to suppress and/or avoid experiences of vulnerability and negative emotions (e.g., sadness) leads to the proliferation of these emotions over time. This may manifest in externalising depression symptoms, leading to negative effects in the form of suicidal/self-harm ideation. In clinical settings, the common alignment of traditional masculine socialization with avoidant coping processes [39,62] and male-type depressive symptoms [63] can both be misinterpreted as normative masculine behaviour (i.e., ‘men being men’), resulting in underlying depression being overlooked.

Avoidant coping may have been more common in the context of the pandemic, where traditional effective avenues of coping (e.g., enlistment of social support) were often cut off due to social distancing measures. Alongside this, there may have been a sense of defeat or helplessness experienced by individuals given the uncontrollable nature of the pandemic as a source of stress. Indeed, recent evidence supports the commonality of avoidance-oriented coping strategies (e.g., substance use and self-distraction) among men in response to distress induced during the pandemic [49,64]. However, in the same research, a broad range of healthy adaptation-oriented coping strategies (e.g., acceptance of uncontrollable pandemic-related stressors, positive reframing) were also reported, suggesting the capacity for effective coping even among more traditionally masculine-identifying men [49,64]. As highlighted in recent empirical findings surrounding help-seeking, whereby help-seeking is framed as indicative of strength and self-betterment when serving the wellbeing of those closest to men [65,66,67], these adaptive coping strategies can then be amplified. In this way, whilst avoidant coping might come more naturally to *some* men, public health promotion efforts should aim to leverage strength-based masculinities that can incorporate effective coping profiles as indicative of masculine self-betterment [68]. For example, recent work has aimed to promote help-seeking in the context of suicide risk among young men as an adaptive coping measure, by directly aligning such self-betterment activities with masculine strength [69]; challenging the notion that seeking help and admitting one’s vulnerabilities is indicative of weakness according to masculine norms [39].

The present study highlights the utility of the MDRS-7 as a screening tool for depression in men and reflects certain advantages over both the Brief COPE and existing screening tools such as the PHQ-9 [27]. The MDRS-7 is much shorter in length compared to the full Brief COPE scale (7 and 28 items, respectively), rendering it far more practical as a clinical screening tool. Nevertheless, if using the MDRS-7 as a screening tool, its symbiotic relationship with avoidant coping (e.g., shared items measuring alcohol abuse) must be kept in mind. In comparison to the PHQ-9, the MDRS-7 screens for symptoms such as emotional suppression and drug use in men, both of which predicted recent suicidal/self-harm ideation in this study. Use of the MDRS-7 (e.g., if delivered in general practice) may therefore help identify cases of men at risk of suicide who would otherwise go undetected. The growing evidence for the MDRS as a reliable and valid indicator of underlying depression [34,35,70] and risk of suicide in men [32] along with its increased ease of administration due to the shortened length of the scale, warrants suggestion for its use in primary care settings to screen for depression in men. 

While the present study has advanced understandings of the role of male-type depressive symptoms in the relationship between avoidant coping and suicidal/self-harm ideation, some limitations must be kept in mind. We did not include a measure of adherence to masculine norms in our analysis, meaning we could not confirm the role of masculine socialization in contributing to avoidant coping and male-type depressive symptoms, though such relationships have been reported elsewhere [22,31]. Additionally, we did not analyse the relationship of other coping strategies (e.g., active strategies such as positive reframing) to suicidal/self-harm ideation. Recent research identified a group of men who used a mix of avoidance and active strategies during the pandemic, who, despite using active coping strategies that were associated with lower psychopathology in other groups, reported more severe stress, depression, and anxiety [49]. It is therefore important to note that avoidant coping is but one part of the story surrounding links between male-type depression and suicidality in men. Further exploring relationships between typically effective or ineffective coping strategies, men’s depression and suicidality is an important area of future research. Additionally, we did not ask participants to report their cultural or ethnic background, meaning we could not assess the role of these factors in contributing to avoidant coping and male-type depressive symptoms. Finally, we used a single item (PHQ-9 item 9) to measure suicidal/self-harm ideation, meaning we could not distinguish between these two outcomes. 

This study provided further evidence in favour of employing the MDRS-7 as a brief screening measure for depression in men. Future research should employ a longitudinal design to investigate the relationship between the variables studied here over time. Given the mediating role of male-type depressive symptoms between avoidant coping and suicidal/self-harm ideation, future research could also investigate coping strategies and strengths-based approaches as intervention targets for depressed men presenting with externalizing symptoms, in order to reduce their risk of suicide and self-harm. Finally, we call for well-designed studies to address broader unanswered questions related to the male-type depression phenotype [71]. Given DSM-5-TR [72] now references men’s externalizing symptoms in a new accompanying statement related to sex- and gender-related diagnostic issues for major depressive disorder, better understanding the complex and nuanced symptomatic picture of men’s depression remains an important task for not just men’s depression assessment, but for improving the fit and effectiveness of treatments offered to them.

## 5. Conclusions

Avoidant coping is associated with suicidal/self-harm ideation in men, and this relationship is mediated by male-type depressive symptoms. Male-type depressive symptoms should therefore be screened for in men presenting to primary care settings with mental health problems. Encouraging depressed men to utilize effective coping strategies may reduce their risk of suicide and/or self-harm.

## Figures and Tables

**Table 1 ijerph-19-10874-t001:** Standardised CFA loadings for MDRS-7 items (*n* = 709).

MDRS-7 Item	Coef	*SE*	95% CI	Z, *p*
1. I bottled up my negative feelings	0.61	0.03	0.55–0.67	20.0, <**0.001**
2. I needed alcohol to help me unwind	0.56	0.03	0.49–0.62	17.0, <**0.001**
3. I had unexplained aches and pains	0.38	0.04	0.30–0.45	9.90, <**0.001**
4. I overreacted to situations with aggressive behaviour	0.66	0.03	0.60–0.72	22.5, <**0.001**
5. It was difficult to manage my anger	0.45	0.03	0.38–0.52	12.5, <**0.001**
6. Using drugs provided temporary relief	0.66	0.03	0.60–0.72	22.8, <**0.001**
7. I stopped caring about the consequences of my actions	0.68	0.03	0.62–0.74	23.6, <**0.001**

Note. Boldface text indicates statistically significant values at *p* < 0.01, Coef = model coefficient, *SE* = standard error, CI = confidence interval.

**Table 2 ijerph-19-10874-t002:** Sample means, SDs and intercorrelations among study variables.

	Mean	SD	*n*	α	2.	3.	4.
1. MDRS-7	9.43	5.75	709	0.78	0.63 ***	−0.36 ***	0.44 ***
2. Avoidant Coping	15.34	4.29	635	0.67		−0.38 ***	0.45 ***
3. Resilience	3.21	0.97	611	0.87			−0.38 ***
4. Suicide/self-harm risk	0.50	0.86	684	-			

Note. *** indicates significance at *p* < 0.001.

**Table 3 ijerph-19-10874-t003:** Responses to COVID-19 items (*n* = 606).

	Response, % (*n*)				
How have government COVID-19 restrictions affected your mental health?	*No stress* 41.9 (254)	*A little* 22.0 (133)	*Moderate* 14.4 (87)	*Considerable* 12.7 (77)	*Extreme* 9.1 (55)
How have government COVID-19 restrictions affected your mental health?	*Very positively* 3.1 (19)	*Somewhat positively* 6.3 (38)	*Not at all* 23.9 (145)	*Somewhat negatively* 43.2 (262)	*Very negatively* 23.4 (142)

**Table 4 ijerph-19-10874-t004:** Mediation model assessing the mediating role of male-type depression in the relationship between avoidant coping and suicidal/self-harm ideation (*n* = 606).

	*β*	*B* (*SE*)	99% CI	T, *p*
Direct effect of AC on SI/SHI	0.20	0.04 (0.01)	0.02–0.07	4.325, <**0.001**
Total effect of AC on SI/SHI	0.31	0.06 (0.01)	0.04–0.09	7.828, <**0.001**
Indirect effect of AC on SI/SHI (via MDRS-7 score)	0.11	0.02 (0.01)	0.01–0.04	
**Covariates**				
Age	0.05	0.003 (0.002)	−0.002–0.008	1.399, 0.162
Resilience	−0.20	−0.17 (0.04)	−0.27–0.09	−5.062, <**0**.**001**
COVID: financial stress	0.09	0.06 (0.02)	0.00–0.121	2.461, 0.014
COVID: government restrictions impact	0.05	0.05 (0.03)	−0.03–0.13	1.434, 0.152

Note. Boldface text indicates statistically significant values at *p* < 0.01, *β* = standardised coefficient, *B* = unstandardised coefficient, AC = avoidant coping, SI/SHI = suicidal ideation/self-harm ideation, MDRS-7 = masculine depression risk scale, CI = confidence interval, *SE* = standard error.

**Table 5 ijerph-19-10874-t005:** MDRS-7 items predicting recent suicide and self-harm ideation (*n* = 684).

MDRS-7 Item	Coeff	AOR	95% CI	*SE*	*p*
1. I bottled up my negative feelings	0.53	1.70	1.40–2.07	0.17	**<0.001**
2. I needed alcohol to help me unwind	0.11	1.12	0.96–1.30	0.09	0.157
3. I had unexplained aches and pains	0.19	1.02	0.89–1.16	0.07	0.786
4. I overreacted to situations with aggressive behaviour	−0.07	0.92	0.74–1.17	0.11	0.532
5. It was difficult to manage my anger	0.20	1.22	1.03–1.44	0.10	**0.016**
6. Using drugs provided temporary relief	0.57	1.77	1.47–2.13	0.17	**<0.001**
7. I stopped caring about the consequences of my actions	0.09	1.10	0.86–1.37	0.13	0.456

Note. Boldface text indicates statistically significant values at *p* < 0.05.

## Data Availability

Data requests will be considered on a case-by-case basis, provided that the requesting researcher has gained ethics approval from a human research ethics committee. Should a request be approved, all data provided will be de-identified.

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
