# Peer review of "Male-Type Depression Mediates the Relationship between Avoidant Coping and Suicidal Ideation in Men"

_ijerph, 2022, doi:10.3390/ijerph191710874_

Round 1
Reviewer 1 Report
Thank you for the opportunity to review this very interesting paper on such an important topic. Excellent background was provided. The authors make a very compelling case.
METHODS
1. Beyond Aboriginal or Torres Strait Islander or not – was any other racial/cultural background information collected?
DISCUSSION
1. Could further explanation be offered regarding “dual copers”? Does this mean dual copers were at greater risk than those who only employed avoidant coping strategies? Why is that?
2. Since masculinity is socioculturally defined, it would have been good to see further measures of racial/cultural background included in the survey. This can be included in the limitations.
3. Could some practical examples of initiatives that “leverage strength-based masculinities” be offered?
4. The authors provide compelling evidence to support to use of the MDRS-7
5. Are there mental health promotion programs that have been shown to be effective in helping men learn to use better coping methods? What is the best way to offer this type of intervention?
Reviewer 2 Report
Thanks for inviting me to review this paper. This paper found that “male-type depression symptoms to better identify men at risk of suicidal/self-harm ideation.” I have the following comments for the authors to address.
1. For the introduction, the authors should highlight why there is a need to study male depression. There is no reference to male homosexuals and challenges faced by them. I recommend the authors to search PubMed for “Depression and male homosexuals [ti]” and select some studies to highlight in the introduction and justify for this study.
2. Under the method, please state that “Male depressive symptoms were assessed via the recently-validated MDRS-7 [36] in Australia” Please add “in Australia”.
3. For the results, first paragraph, it would be good if the authors can specify the prevalence of alcohol dependence, substance misuse, past psychiatric history, antidepressant treatment, psychotherapy for depression and forensic history in the first paragraph. If such information are not available, please state as a limitation.
4. This study focused on using questionnaire to identify male-type of depression. The authors should discuss recent advances in functional neuroimaging to assess depressive disorder. For example, there is a research finding in the PubMed that found "75.2% and 76.5% of patients with MDD were correctly classified using frontal and temporal region oxy-haemoglobin, respectively". Would there be a difference between men and women? Please discuss.
5. This study only used 2 questions to screen for impact of COVID-19 and it is too brief. Please refer to other questionnaires and stated the items that were missing and not assessed in this study.
